# Chemical Modification of Auranofin Yields a New Family of Anticancer Drug Candidates: The Gold(I) Phosphite Analogues

**DOI:** 10.3390/molecules28031050

**Published:** 2023-01-20

**Authors:** Damiano Cirri, Andrea Geri, Lara Massai, Michele Mannelli, Tania Gamberi, Francesca Magherini, Matteo Becatti, Chiara Gabbiani, Alessandro Pratesi, Luigi Messori

**Affiliations:** 1Department of Chemistry and Industrial Chemistry, University of Pisa, Via G. Moruzzi 13, 56124 Pisa, Italy; 2Laboratory of Metals in Medicine (MetMed), Department of Chemistry “Ugo Schiff”, University of Florence, Via della Lastruccia 3-13, 50019 Sesto Fiorentino, Italy; 3Department of Experimental and Clinical Biomedical Sciences “Mario Serio”, University of Florence, Viale G.B. Morgagni 50, 50134 Firenze, Italy

**Keywords:** auranofin, metal-based drugs, anticancer compounds, phosphite compounds

## Abstract

A panel of four novel gold(I) complexes, inspired by the clinically established gold drug auranofin (1-Thio-β-D-glucopyranosatotriethylphosphine gold-2,3,4,6-tetraacetate), was prepared and characterized. All these compounds feature the replacement of the triethylphosphine ligand of the parent compound auranofin with a trimethylphosphite ligand. The linear coordination around the gold(I) center is completed by Cl^−^, Br^−^, I^−^ or by the thioglucose tetraacetate ligand (SAtg). The in-solution behavior of these gold compounds as well as their interactions with some representative model proteins were comparatively analyzed through ^31^PNMR and ESI-MS measurements. Notably, all panel compounds turned out to be stable in aqueous media, but significant differences with respect to auranofin were disclosed in their interactions with a few leading proteins. In addition, the cytotoxic effects produced by the panel compounds toward A2780, A2780R and SKOV-3 ovarian cancer cells were quantitated and found to be in the low micromolar range, since the IC_50_ of all compounds was found to be between 1 μM and 10 μM. Notably, these novel gold complexes showed large and similar inhibition capabilities towards the key enzyme thioredoxin reductase, again comparable to those of auranofin. The implications of these results for the discovery of new and effective gold-based anticancer agents are discussed.

## 1. Introduction

Auranofin (1-Thio-β-D-glucopyranosatotriethylphosphine gold-2,3,4,6-tetraacetate) (AF hereafter) is a gold drug (Figure 1) approved by the FDA, in 1985, for the treatment of some severe forms of rheumatoid arthritis (RA). AF consists of a gold(I) center linearly coordinated to a triethylphosphine and a tetracetylthioglucose ligand. AF behaves as a typical prodrug requiring chemical activation to perform its biological actions; the first step of the activation process is the release of the thiosugar moiety. AF, once activated, can bind tightly to a variety of cellular proteins, in particular proteins containing free cysteines, thus producing its cellular effects. Notably, in the frame of the so-called “drug repurposing” approach, AF has been tested for the treatment of many other clinical indications beyond RA, ranging from parasitic to neoplastic diseases, with promising results. In particular, AF gained great attention in the field of metal-based anticancer drugs owing to the encouraging results obtained from a variety of tumoral models [1]. Moreover, AF is currently in clinical trials for the treatment of non-small cell lung cancer and ovarian cancer, with some published results regarding this latter disease [2]. For these reasons, AF is currently considered the reference compound for the emerging class of gold-based anticancer drugs and should come as no surprise, considering the wide literature reports on many AF modifications.

Due to the biomedical relevance of AF, it is reasonable to try to modify its structure systematically in the search for new chemical entities endowed with a better pharmaceutical profile. Accordingly, several efforts have been conducted to make small and systematic chemical modifications to the AF scaffold, with the aim of improving its anticancer properties. Many modification attempts were described so far, such as the preparation of derivatives in which the thiosugar ligand is replaced by halide ligands [3] or other biologically active molecules [4]. Another relevant modification was reported by F. Shaw who replaced the sulfur of the thiosugar ligand with a selenium atom, affording the so-called selenoauranofin [5]; other studies explored the effects of a variety of modifications in the phosphine ligand [6,7]. Here, we analyze the effects of the replacement of the triethylphosphine group with a trimethylphosphite ester. The choice of trimethylphosphite is justified by its isoelectronic structure with respect to triethylphosphine combined with a lower lipophilicity that should reduce the typical solubility issues related to AF and its derivatives [8]. Additionally, the anionic ligand has systematically been varied starting from the classical thioglucose tetraacetate and moving down in the halogen group (Cl, Br and I). The resulting four compounds, depicted in Figure 2, were prepared, characterized, and then evaluated for their in-solution behavior. More precisely, the chemical stability of the whole panel of novel gold compounds was evaluated in aqueous media through ^31^PNMR, while their reactivity patterns towards two representative biomolecules, i.e., human carbonic anhydrase I (hCA I) and human serum albumin (HSA), were assessed through ESI-MS. To better understand the effects of oxygen introduction on the hydrophilicity of these gold compounds, a comparative LogP and solubility analysis was carried out. Finally, the in vitro cytotoxic activities of the synthesized compounds were evaluated in comparison with AF against the A2780 ovarian cancer model, in connection to TrxR inhibition experiments.

On the whole, the novel compounds revealed some interesting features, such as good binding properties towards selected biological targets as well as the maintenance of in vitro cytotoxicity associated with better water solubility than AF. Additionally, the inhibition experiments towards thioredoxin reductase produced encouraging results, with the determined IC_50_ values being in the same range as those reported for AF.

## 2. Results and Discussion

### 2.1. Chemistry

The first gold(I) complex in this series, i.e., AuP(OCH_3_)_3_Cl, was obtained through a simple ligand exchange reaction, in which the commercially available chloro(dimethylsulfide)gold(I) complex was reacted with a stoichiometric amount of trimethylphosphite. The reaction produced AuP(OCH_3_)_3_Cl in an almost quantitative yield (95%), and its ^31^PNMR spectrum was fully consistent with the data already reported in the literature [9]. In turn, AuP(OCH_3_)_3_Cl was employed as the starting material for the preparation of the other panel compounds. Indeed, AuP(OCH_3_)_3_Br, AuP(OCH_3_)_3_I, and AuP(OCH_3_)_3_SAtg (SAtg = 1-Thio-β-D-glucose tetraacetate) were obtained through the chloride anion displacement reaction, as described in the experimental section. Interestingly, the synthesized compounds show LogP values that, due to the presence of three oxygen atoms on the phosphorus ligand, turned out to be far lower with respect to their parent compounds. Indeed, AF, Au(PEt_3_)Cl, and Au(PEt_3_)I are reported to have LogP values of 1.6, 1.7 and 4.6, respectively [10]. Conversely, as described in the experimental section, the LogP values of the synthesized compounds ranged between −0.7 and 0.7. This difference should result, at least in principle, in better water solubility. For this reason, we determined the solubilities of all compounds through a modification of the experimental protocol reported in [11] (details are discussed in the “Materials and Methods” section). Notably, all investigated compounds turned out to be far more soluble in water than the parent triethylphosphine gold(I) complexes (Table 1).

This structural modification might also affect other chemical properties that in turn may lead to significant differences in the respective biological profiles. For this reason, we first determined the solution stability of the whole panel compounds through ^31^PNMR spectroscopy. As reported in the Appendix A, the investigated compounds turned out to be fully stable over 72 h in a 1:1 methanol/water solution.

### 2.2. Protein Interactions

The panel compounds were then challenged with the two following proteins: HSA and hCA I. Both selected biomolecules contain a free cysteine residue which is, according to several published papers, a preferential binding site for gold(I) complexes due to the high affinity of gold(I) for amino acids residues bearing sulfur atoms [12,13,14]. Thus, the interactions between all study compounds and the mentioned proteins were investigated through ESI-MS experiments according to a well-established experimental protocol [4,15]; indeed, in the recent literature, several studies characterizing the formation of adducts between a variety of model proteins and many metal-based drugs have been reported [16,17]. The recorded ESI mass spectra highlight that all gold compounds act as prodrugs by binding the proteins through a metal fragment, which is the “active part” of the compound itself (Figure 3 and Figure 4). The deconvoluted mass spectrum of HSA (Figure 3A) shows the main signal at 66,437.379 Da assigned to the native protein, while the signal at 66,556.628 Da is attributed to protein cysteinylation, which is a typical post-translational modification of serum albumin. The deconvoluted mass spectrum of free hCA I is characterized by the signal at 28,843.328 Da attributed to the holo-protein (Figure 4A). Upon reaction, all complexes manifest a good reactivity with both proteins giving rise to a signal at a greater mass value than the free protein, mainly related to the addition of the cationic metal fragment [AuP(OCH_3_)_3_]^+^. Conversely, AuP(OCH_3_)_3_Br and AuP(OCH_3_)_3_I interact with HSA giving an adduct only bearing one bound gold(I) ion, having lost all the original ligands. The persistence of the signal of the free protein in the spectra indicates that the metalation of the protein is not yet complete after 24 h of incubation.

### 2.3. Biological Evaluations

Subsequently, the panel compounds were investigated for their in vitro anticancer properties. First, as reported in Table 2, their cytotoxicity was evaluated against three different ovarian cancer cell lines, i.e., A2780, cisplatin-resistant A2780 (A2780R) and SKOV-3. Notably, all the synthesized compounds showed IC_50_ values falling in the low micromolar region, with the AuP(OCH_3_)_3_SAtg compound being the most performing one against both A2780 and A2780R. Conversely, the iodide derivative, AuP(OCH_3_)_3_I, turned out to be the most active compound for SKOV-3 cells. We chose these cell lines since they have been already used in our previous studies as cancer model cells. Moreover, all panel compounds showed outstanding activity towards the cisplatin-resistant A2780R cell line. In particular, AuP(OCH_3_)_3_SAtg showed an activity nearly superimposable to that produced in the A2780 line. This latter experimental evidence opens very promising opportunities in the frame of the well-known ineffectiveness of cisplatin in the treatment of resistant ovarian cancer models such as A2780R [18].

The roughly similar cytotoxic activity determined for the various gold compounds in A2780 cells is supported by the analytical determinations of the metal content. Indeed, in order to evaluate the gold accumulation in A2780 ovarian cancer cells, metal uptake studies were performed by ICP-OES; the obtained results demonstrate that a substantially comparable amount of the metal was accumulated in the cells treated with the four different gold compounds (Figure 5); just as in the case of AuP(OCH_3_)_3_Cl, a somewhat greater gold uptake was measured.

Caspase-3 belongs to the family of the so-called “executive caspases” and is expressed as an inactive zymogen; upon activation by both the intrinsic and extrinsic death pathways, it induces typical apoptotic events such as membrane blebbing and DNA fragmentation [19,20]. To investigate if the cytotoxic effects elicited by gold compounds are mediated by apoptotic cell death, we evaluated the caspase-3 activation by flow cytometry. Figure 6 shows the percentage of cells with activated caspase-3.

All compounds cause a statistically significant activation of this caspase indicating that their cytotoxicity leads to cell death through the apoptotic pathway.

To verify whether the enzyme thioredoxin reductase is also a likely target for these novel gold compounds as it is for AF, we measured the TrxR activity in A2780 cells after drug exposure. The TrxR enzyme activity was quantified through the measurement of the colourimetric reduction of DTNB into TNB at 412 nm. The results show that all the investigated AF analogues markedly inhibit TrxR enzyme activity, as already demonstrated for auranofin (more precisely the latter was found to cause 40–50% inhibition of TrxR activity [3]) (Figure 7).

Even if the differences between the four AF analogues were not statistically significant, AuP(OCH_3_)_3_SAtg showed a greater inhibitory potency being able to reduce the TrxR to about 55% with respect to controls, followed -in decreasing order of inhibition- by AuP(OCH_3_)_3_Cl (62%), AuP(OCH_3_)_3_I (68%) and AuP(OCH_3_)_3_Br (73%).

## 3. Materials and Methods

### 3.1. Synthesis

AuP(OCH_3_)_3_Cl: 315 mg of Au(SMe_2_)Cl (1.07 mmol) were added in a 50 mL flask and solubilized with 20 mL of dichloromethane. Next, 126 μL (1.07 mmol) of trimethylphosphite were added under stirring. After 30 min at room temperature, the solution was filtered, and the solvent removed under vacuum. The wet white product was dried affording 364 mg of a crystalline white solid (yield 95%).

Elemental analysis: calc. C 10.11%, H 2.54%; exp. C 10.37%, H 2.32%

^1^HNMR (400 MHz; CDCl_3_): 3.78 (d, 3H, *J* = 13.96 Hz)

^13^CNMR (100 MHz; CDCl_3_): 53.9

^31^PNMR (160MHZ; CDCl_3_): 120.7

LogP: 0.6

AuP(OCH_3_)_3_Br: 104 mg of AuP(OCH_3_)_3_Cl (0.29 mmol) were added in a 50 mL flask and solubilized with 20 mL of ethanol. Next, 176 mg of KBr (1.45 mmol) were added to the solution and the suspension was stirred at r.t. for 1 h. Subsequently, the solvent was removed under vacuum and the crude product was suspended in chloroform. The suspension was filtered to remove the residual salts and the solution was concentrated through a rotary evaporator until the appearance of a viscous product. Subsequently, the pure product was precipitated through the addition of hexane. The product was recovered as 82 mg of white solid through Buchner funnel filtration (yield 70%).

Elemental analysis: calc. C 8.99%, H 2.26%; exp. C 9.23%, H 1.98%

^1^HNMR (400 MHz; CDCl_3_): 3.74 (d, 3H, *J* = 13.96 Hz)

^13^CNMR (100 MHz; CDCl_3_): 53.7

^31^PNMR (160MHZ; CDCl_3_): 123.9

LogP: 0.1

AuP(OCH_3_)_3_I: 103 mg of AuP(OCH_3_)_3_Cl (0.29 mmol) were added in a 50 mL flask and solubilized with 20 mL of ethanol. 241 mg of KI (1.46 mmol) were added, and the suspension was stirred at r.t. for 2 h. Subsequently, the solvent was removed through a rotary evaporator and the crude product was suspended in chloroform. The suspension was filtered to remove the residual salts, and the solution was concentrated through a rotary evaporator until the appearance of a viscous product. Subsequently, hexane was added, and the flask was kept at −20 °C overnight. The product was recovered as 62 mg of white-violet solid through Buchner funnel filtration (yield 48%).

Elemental analysis: calc. C 8.04%, H 2.03%; exp. C 8.24%, H 1.92%

^1^HNMR (400 MHz; CDCl_3_): 3.81 (d, 3H, *J* = 14.00 Hz)

^13^CNMR (100 MHz; CDCl_3_): 53.6

^31^PNMR (160MHZ; CDCl_3_): 132.4

LogP: 0.7

AuP(OCH_3_)_3_SAtg: 49 mg of AuP(OCH_3_)_3_Cl (0.14 mmol) were added in a 25 mL flask together with 52 mg of 1-thio-β-D-glucose tetraacetate (0.14 mmol) and solubilized with 7 mL of ethanol. 30 mg of NaHCO_3_ (3.53 mmol) were added to the solution. Then, the suspension was stirred a r.t. for 4 h. Subsequently, the solvent was removed through a rotary evaporator and the crude product was suspended in chloroform. The suspension was filtered to remove the residual salts and the solution was concentrated through a rotary evaporator until the appearance of a viscous product. Subsequently, hexane was added, and the flask was kept at −20 °C overnight. The product was recovered through decantation as 76 mg of a white sticky solid (yield 80%).

Elemental analysis: calc. C 29.83%, H 4.12 %, S 4.68%; exp. C 30.21%, H 4.08%, S 4.62%

^1^HNMR (400 MHz; CDCl_3_): 5.12 (m, 3H); 4.98 (m, 1H); 4.24 (m, 1H); 4.11 (m, 1H); 3.78 (d, 3H, *J* = 13.70 Hz); 3.73 (s, 1H); 2.08 (s, 3H); 2.06 (s, 3H); 2.01 (s, 3H); 1.98 (s, 3H)

^13^CNMR (100 MHz; CDCl_3_): 83.6; 78.0; 76.5; 74.8; 69.4; 63.3; 62.7; 54.4; 21.7; 21.3

^31^PNMR (160MHZ; CDCl_3_): 138.2

LogP: −0.7

### 3.2. NMR Experiments

All NMR spectra were acquired on a JEOL 400YH spectrometer (resonating frequencies: 400, 160 and 100 MHz for ^1^H, ^31^P and ^13^C, respectively). All spectra were recorded at room temperature (25 ± 2 °C) in solvents with a deuteration degree of 99.8% and calibrated on solvent residual signals [21]. All deuterated solvents were purchased from Deutero.de (https://www.deutero.de/, accessed on 17 December 2022). ^1^H, ^31^P and ^13^C characterization spectra were recorded in CDCl_3_. Conversely, for performing solution stability studies, a small amount of each compound has been solubilized in 0.25 mL of MeOD-d_4_. Subsequently, 0.25 mL of H_2_O were added to each sample and the phosphorus signal was monitored up to 72 h for confirming the stability in aqueous media.

### 3.3. Solubility and LogP Determination Protocols

For determining the solubility of investigated compounds, 5 mg of the selected gold(I) complex was suspended in an Eppendorf test tube with 300 μL of D_2_O and 300 μL of a Me_2_SO_2_/D_2_O solution 6.94 mM (Me_2_SO_2_ final conc. 3.47 mM). The suspension was heated to 40 °C and sonicated for 90 min. The resulting saturated solution was left to cool to room temperature, decanted, transferred to an NMR tube and then analyzed through quantitative ^1^HNMR spectroscopy (tilt angle = 45°; recycle delay = 4 s; number of scans = 16). The concentration (solubility) was calculated by the relative integral (related to phosphine or phosphite -CH_3_ groups) with respect to Me_2_SO_2_ singlet (δ = 3.14 ppm). LogP values were determined through a modification of the shake-flask method reported in [22]. More precisely, in a 15 mL Falcon tube, roughly 2 mg of the sample was added in 2 mL of n-octanol-saturated water and 2 mL of water-saturated n-octanol. The mixture was hand shaken for ten minutes, and then 0.5 mL of each phase were moved in two different mineralization tubes together with 1 mL of metal-free concentrated nitric acid. The mixtures were heated overnight at 90 °C, then diluted with 4.5 mL of ultrapure water. The gold concentration in each sample was determined through ICP-OES and the respective LogP values were calculated through the ratio between octanol and water gold concentrations.

### 3.4. ESI-MS Experiments

The ESI-MS investigations were performed using a TripleTOF^®^ 5600+ high-resolution mass spectrometer (Sciex, Framingham, MA, USA), equipped with a DuoSpray^®^ interface operating with an ESI probe. All the ESI mass spectra were acquired through direct infusion at 7 μL/min flow rate. The ESI source parameters optimized for the proteins are as follows:

HSA: positive polarity, ion-spray voltage floating (ISFV) 5500 V, temperature (TEM) 25 °C, ion source gas 1 (GS1) 40 L/min; ion source gas 2 (GS2) 0 L/min; curtain gas (CUR) 20 L/min, collision energy (CE) 10 V; declustering potential (DP) 200 V, acquisition range 1000–2500 *m*/*z*.

hCA I: positive polarity, ion-spray voltage floating (ISFV) 5500 V, temperature (TEM) 25 °C, ion source gas 1 (GS1) 25 L/min; ion source gas 2 (GS2) 0 L/min; curtain gas (CUR) 20 L/min, collision energy (CE) 10 V; declustering potential (DP) 300 V, acquisition range 1600–3400 *m*/*z*.

For acquisition, Analyst TF software 1.7.1 (Sciex) was used and deconvoluted spectra were obtained by using the Bio Tool Kit micro-application v.2.2 embedded in PeakView™ software v.2.2 (Sciex). All samples were diluted to a final protein concentration of 10^−6^ M using a 20 mM ammonium acetate solution, pH 6.8, and in the samples of HSA 0.1% *v*/*v* of formic acid was added just before the infusion in the mass spectrometer in order to enhance the ionization process.

### 3.5. Cell Lines and Culture Conditions

RPMI 1640 cell culture medium, fetal bovine serum (FBS), and phosphate-buffered saline (PBS) were obtained from Euroclone (Milan, Italy). Thiazolyl blue tetrazolium bromide (MTT) was obtained from Merck Life Science (Milan, Italy). The human ovarian cancer cell lines A2780, A2780/CDDP-R (cisplatin-resistant) and SKOV3 were obtained from ATCC. Cell lines were maintained in RPMI-1640 supplemented with 10% fetal calf serum (FCS) and antibiotics (penicillin, 100 U/mL; streptomycin, 100 μg/mL) at 37 °C in a 5% CO_2_ humidified atmosphere and subcultured twice weekly. Split 1:5 (3–6 × 10^4^ cells per mL).

### 3.6. Cytotoxicity Assessment

The cytotoxic effects of auranofin analogues on ovarian cancer cell lines were determined by using the colourimetric MTT (3-(4,5-dimethylthiazol-2-yl)-2,5-diphenyltetrazolium bromide) assay. Viable cells can metabolize MTT into formazan, a blue-coloured compound that is detectable at 595 nm. Exponentially growing cells were seeded in 96-well plates at a density of 10^4^ cells/well. After 24 h, cells were treated with increasing concentrations of gold compounds ranging from 1 nM to 100 μM and each concentration was tested in triplicate. After 72 h, 0.5 mg/mL of MTT was added to each well and maintained at 37 °C for 1 h. Following precipitation, blue formazan was dissolved in 100 µL of DMSO, and the optical density was read in a microplate reader interfaced with Microplate Manager/PV version 4.0 software (BioRad Laboratories, Hercules, CA, USA). From the absorbance measurements, the half-maximal inhibitory concentration (IC_50_) value of each compound on ovarian cancer cells was calculated using GraphPad Prism software version 6.0 (Graphpad Holdings, LLC, Boston, MA, USA). Each MTT assay was performed at least in triplicate and the results are reported in Table 2 as IC_50_ mean values ± standard deviation (SD).

The effects of these calculated 72 h exposure IC_50_ doses on A2780 cell viability were also evaluated using an MTT time course assay at 24, 48, and 72 h of drug exposure (See Appendix A). The experimental protocol described above was applied.

### 3.7. Thioredoxin Reductase Inhibition

Exponentially growing cells were seeded in 20 mm^2^ Petri dishes at a density of 6 × 10^5^ cells/dish for 24 h. Thereafter, A2780 cells were treated with gold compound concentrations corresponding to their 72-h exposure IC_50_ doses for 24 h. At this time, cells are viable as demonstrated by MTT time course experiments (Appendix A). At treatment, cells were lysed with RIPA buffer (50 mM Tris-HCl pH 7.5, 150 mM NaCl, 100 mM NaF, 2 mM EGTA, 1% Triton X-100) containing 10 μL/mL protease and phosphatase inhibitors (Sigma Aldrich). Lysates were centrifugated at 4 °C, 14,000 RPM for 15 min, and supernatants were collected. After protein quantification with Bradford Assay, 30 μg of proteins were used for the assay. TrxR activity was measured by using a commercial colourimetric assay kit (Sigma Aldrich CS0170) based on the reduction of 5,5′-dithiobis(2-nitrobenzoic acid) (DTNB) with NADPH to 5-thio-2-nitrobenzoic acid (TNB) at 412 nm. This kit also contains a solution of mammalian TrxR inhibitor. Experiments were performed in triplicate. Results were normalized to the cellular protein content and reported as the percentage of enzyme activity compared to the untreated cells (control).

### 3.8. Assessment of Caspase-3 Activity by Flow Cytometry

Caspase-3 activity was analyzed by flow cytometry as previously reported [23]. Briefly, control and treated cells for 72 h were trypsinized, washed with PBS and resuspended in FAM-FLICATM Caspases solution (Caspase FLICA kit FAM-DEVD-FMK, ImmunoChemistry Technologies) for 1 h at 37 °C, following the manufacturer’s instruction. Then, cells were washed twice with PBS, and analyzed by FACSCanto flow cytometer (BD Biosciences, Franklin Lakes, NJ, USA). All the analyses were carried out in triplicate.

### 3.9. Gold Uptake Assessment

To perform the experiments of gold cellular uptake, A2780 cells were plated in 60 mm^2^ Petri dishes at a density of 1.8 × 10^6^ cells per dish in RPMI-1640 complete medium for 24 h. Thereafter, cells were treated for 24 h with gold compound doses corresponding to their 72 h exposure IC_50_ values. The medium was then removed, and cells were collected via trypsinization. Cells were then washed with PBS (1 mL × 3). An aliquot (50 µL) of 1 mL PBS was picked up for the protein amount quantification used to normalize the gold content. The precipitated cellular pellet was mineralized with HNO_3_ to determine the gold content through ICP-OES analysis. The determination of gold concentration in cells was performed by a Varian 720-ES Inductively Coupled Plasma Atomic Emission Spectrometer (ICP-AES) equipped with a CETAC U5000 AT+ ultrasonic nebulizer, in order to increase the method sensitivity. Before the analysis, samples were weighed in PE vials and digested in a thermo-reactor at 80 °C for 8 h with 500 µL of HNO_3_ (35% suprapure grade). After digestion, the samples were diluted to 5.5 mL with ultrapure water (≤18 MΩ). 5.0 mL of each sample were spiked with 1 ppm of Ge used as an internal standard and analyzed. Calibration standards were prepared by gravimetric serial dilution from a commercial standard solution of Au at 1000 mg L^−1^. The wavelength used for Au determination was 267.594 nm, whereas for Ge the line at 209.426 nm was used. The operating conditions were optimized to obtain maximum signal intensity, and between each sample, a rinse solution of HNO_3_ (35% suprapure grade) was used in order to avoid any “memory effect”.

### 3.10. Statistical Analysis

For biological experiments, all values are given as means ± SD of no less than three independent experiments. Statistical analysis was performed by unpaired t-Test or one-way ANOVA test followed by Tukey’s multiple comparison test using GraphPad Prism software version 6.0 (Graphpad Holdings, LLC, Boston, MA, USA). A *p*-value ≤ 0.05 was considered statistically significant. For details, see figure legends.

## 4. Conclusions

In conclusion, we have described here a new family of gold(I)-based anticancer candidates inspired to the approved drug auranofin. Notably, the replacement of triethylphosphine by a more polar trimethylphosphite ester group did not affect substantially the binding properties of the novel compounds towards two representative proteins nor their cytotoxic capabilities. More precisely, the investigated compounds effectively interacted with both HSA and hCA I, while their cytotoxic properties against three ovarian cancer cell lines turned out to fall in the low micromolar range. In addition, we demonstrated that the study compounds, similarly to AF, produced a potent inhibition of the enzyme thioredoxin reductase as well as a clear apoptotic induction, as indicated by caspase-3 activation. Furthermore, the biological evaluations and the solubility data are in nice agreement with the metal uptake studies, which demonstrated that the gold content in the cells is similar for all compounds. Yet, we believe that the appreciable chemical differences that were documented with respect to the parent phosphine compounds, such as the far lower lipophilicity and the greater water solubility, may lead to relevant differences in the overall pharmacokinetic profiles and may also facilitate future pharmaceutical use.

## Figures and Tables

**Figure 1 molecules-28-01050-f001:**
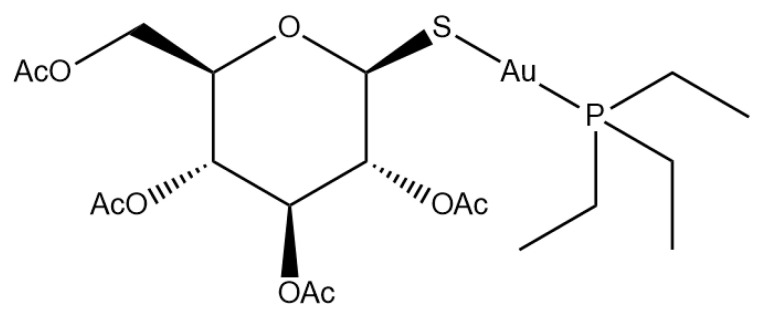
Auranofin structure.

**Figure 2 molecules-28-01050-f002:**
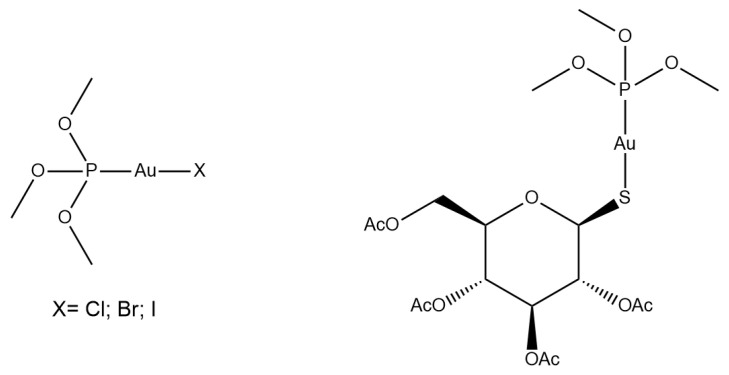
The investigated compounds.

**Figure 3 molecules-28-01050-f003:**
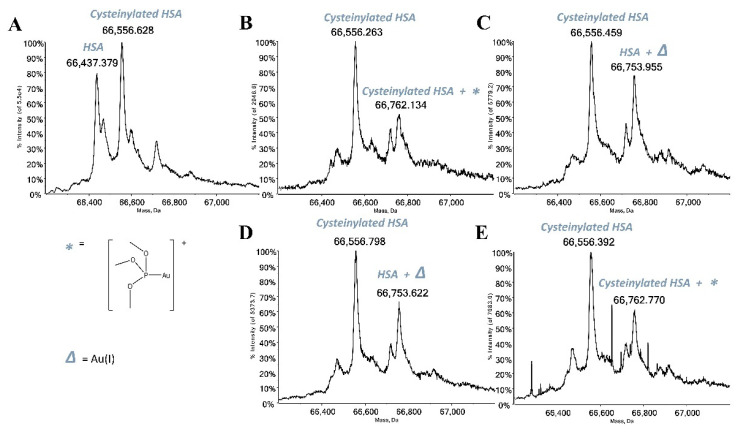
(**A**) Deconvoluted ESI-MS mass spectra of human serum albumin (HSA) 10^−6^ M in 20 mM ammonium acetate solution at pH 6.8, incubated at 37 °C for 5 min with (**B**) AuP(OCH_3_)_3_Cl, (**C**) AuP(OCH_3_)_3_Br, (**D**) AuP(OCH_3_)_3_I, (**E**) AuP(OCH_3_)_3_SAtg solution in a 1:3 protein-to-gold ratio; 0.1% *v*/*v* of formic acid was added just before injection.

**Figure 4 molecules-28-01050-f004:**
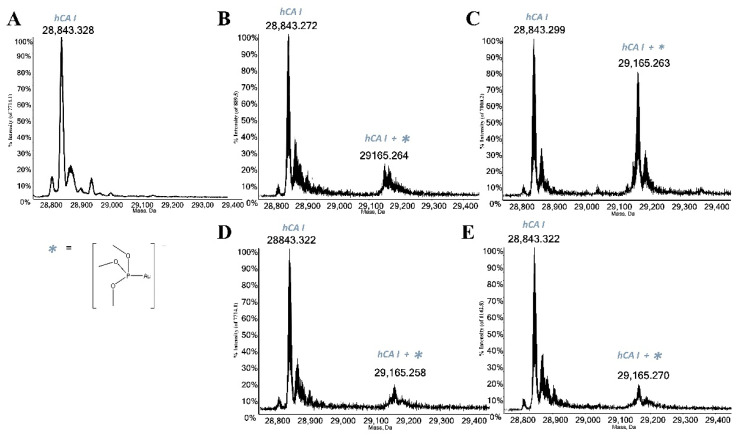
(**A**) Deconvoluted ESI-MS mass spectra of human carbonic anhydrase I (hCA I) 10^−6^ M in 20 mM ammonium acetate solution at pH 6.8 and incubated at 37 °C for 24 h with (**B**) AuP(OCH_3_)_3_Cl, (**C**) AuP(OCH_3_)_3_Br, (**D**) AuP(OCH_3_)_3_I, (**E**) AuP(OCH_3_)_3_SAtg solution in a 1:1 protein-to-gold ratio.

**Figure 5 molecules-28-01050-f005:**
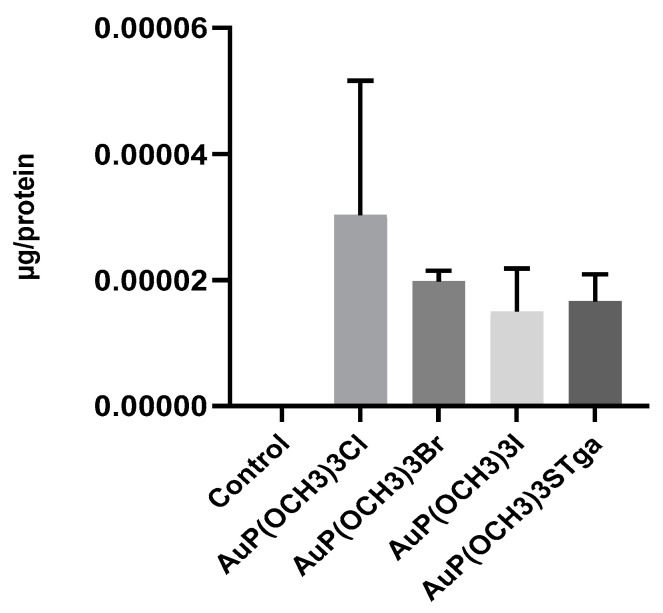
Analysis of gold content in A2780 cells following treatment; the gold content is indicated as a ratio between µg of gold and total protein amount. Histograms report the mean values ± SD.

**Figure 6 molecules-28-01050-f006:**
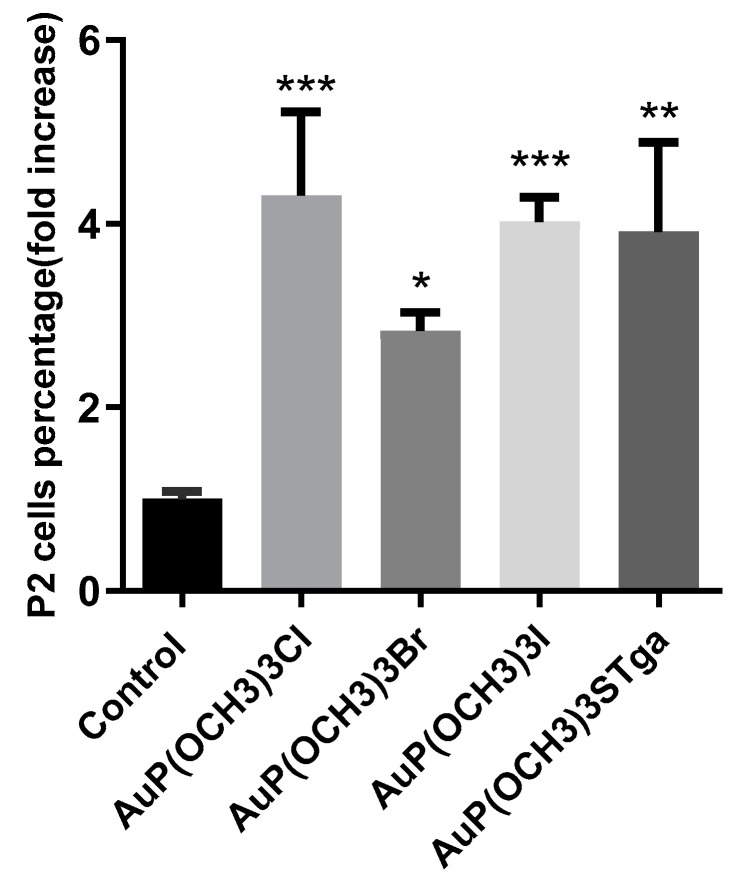
Caspase-3 activity shown by fluorescence-activated cell sorting analysis using FAM FLICA in A2780 cells treated for 72 h with gold compounds concentration corresponding to their 72 h exposure IC_50_-dose. Histograms report the mean values ± SD. P2 represents the percentage of the cell population with activated caspase-3 (P2 population). The statistical analysis was carried out using one-way ANOVA test followed by Tuckey’s multiple comparisons test using Graphpad Prism v 6.0 (* *p* < 0.05, ** *p* < 0.01, *** *p* < 0.001).

**Figure 7 molecules-28-01050-f007:**
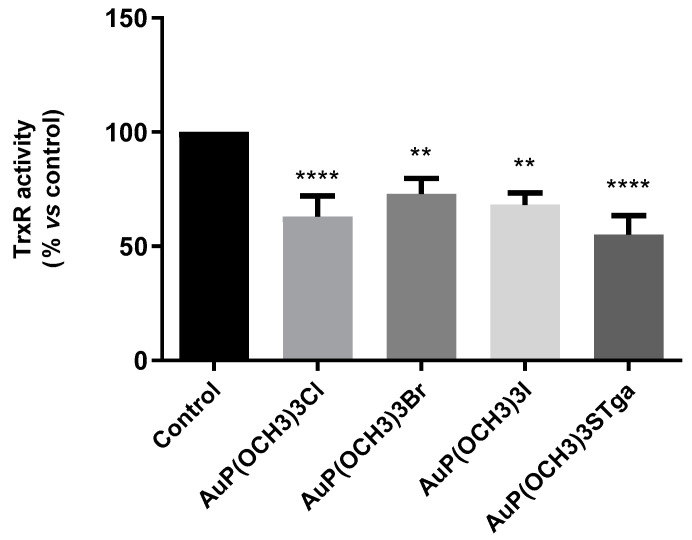
TrxR enzyme inhibition assay was performed after 24 h of treatment using a commercial thioredoxin reductase assay kit, whereby histograms report the residual TrxR enzyme activity in gold-treated cells expressed in percentage with respect to control cells. The analysis was performed in triplicate. The statistical analysis was carried out using one-way ANOVA test followed by Tukey’s multiple comparisons test using Graphpad Prism software v 6.0 (** *p* < 0.01, **** *p* < 0.0001).

**Table 1 molecules-28-01050-t001:** Water solubility of the synthesized compounds in comparison with their parent triethylphosphine derivatives. Data are given as millimolar concentration.

	X = Cl	X = Br	X = I	X = SAtg
Au(PEt_3_)X	0.68	0.14	0.08	0.34
AuP(OCH_3_)_3_X	1.24	0.54	0.19	2.76

**Table 2 molecules-28-01050-t002:** In vitro cytotoxicity of the investigated compounds against A2780, A2780R and SKOV-3 ovarian cancer models. IC_50_ (µM) ^1^ ± SD.

Compound	A2780	A2780R	SKOV-3
AuP(OCH_3_)_3_Cl	2.32 ± 0.25	9.49 ± 1.56	2.39 ± 0.44
AuP(OCH_3_)_3_Br	2.63 ± 0.39	8.23 ± 1.50	4.57 ± 0.10
AuP(OCH_3_)_3_I	2.93 ± 0.46	7.81 ± 1.28	0.99 ± 0.19
AuP(OCH_3_)_3_SAtg	1.26 ± 0.22	2.98 ± 0.23	2.44 ± 0.32

^1^ Half-maximal inhibitory concentration (IC_50_) values of the investigated gold compounds after 72 h of treatment using MTT assay. Values are means ± standard deviation (SD) of three independent experiments.

## Data Availability

All data are contained within the present article and its Appendix A.

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
