# Peer review of "Chemical Modification of Auranofin Yields a New Family of Anticancer Drug Candidates: The Gold(I) Phosphite Analogues"

_molecules, 2023, doi:10.3390/molecules28031050_

Round 1
Reviewer 1 Report
The manuscript by Messori et al. addresses the importance of chemical modification of auranofin in cancer treatment. The authors used different halide substituents in addition to thioglucose coordinated to phosphite ligands. The linear coordination on gold(I) resulted in compounds analogue to auranofin. Some corrections are included below:
1. Abstract and title mentioned "auranofin". However, the synthesized compounds are auranofin analogues. Please correct. Auranofin contains phosphine, however, the work by Messori addresses phosphite. The Bruces from The University of Maine used trimethylphosphine instead of triethylphosphine and called their compound analogue.
2. In several places including the abstract, the authors used both compounds and complexes. Please be consistent and use either compounds or complexes.
3. The abstract is missing the results such as phosphorus NMR values. MTT and IC50 values are not stated. The abstract can include more specific results. Protein interaction studies are not mentioned in the abstract.
4. Keywords are missing phosphites.
5. Please include the paper by the Bruces on auranofin analogue.
https://doi.org/10.1021/ic026057z
6. Authors must discuss the possibility of formation of linear structures such as {P-Au-P]+ or tetrahedral. Please compare your phosphorous NMR with the literature to justify.
7. The comparison with related gold-phosphite structures is missing in the introduction. Authors studied phosphite ligands but did not justify their choice.
8. Line 87" It is stated good yield. Please write the yields and comment.
9. Authors called phosphite ligands "oxygen rich". Why do they call them "rich"? They are simply phosphites and not oxygen rich or poor.
10. Please write the structure of auranofin in the abstract and introduction such as Et3PAu(TATG), TATG = 1-Thio-β-D-glucose tetraacetate. Please check Aldrich. The name and abbreviation of the thioglucose ligand must be corrected and be consistent in the manuscript.
11. The title contains gold but not clear if the authors used gold(I) or gold(III). Please specify that gold(I) was used.
Author Response
see the attache file

Reviewer 2 Report
The manuscript is well written, the experiments were well planned and conducted. I have only some minor comments.
- Emphasise the novelty of the study please.
- Why did the authors choose such and not other cell line for their study? Please provide justification.
- Section 3.5. What was the viability of cells taken to the experiment? How it was tested?
- Line 339: ‘104 cells…’ should be in superscript.
- Figures 5-7 differ from each other in font and graphics. This should be standardized.
- In Methodology there should be section describing statistical analysis.
